# Effect of Specimen Thickness and Stress Intensity Factor Range on Plasticity-Induced Fatigue Crack Closure in A7075-T6 Alloy

**DOI:** 10.3390/ma14030664

**Published:** 2021-01-31

**Authors:** Kenichi Masuda, Sotomi Ishihara, Noriyasu Oguma

**Affiliations:** 1Department of Mechanical Engineering, University of Toyama, Gofuku 3190, Toyama 930-8555, Japan; sotomi.ishihara@gmail.com (S.I.); oguma@eng.u-toyama.ac.jp (N.O.); 2National Institute of Technology, Toyama College, Toyama 939-8630, Japan

**Keywords:** fatigue crack growth behavior, aluminum alloy, CT specimen, plasticity-induced fatigue crack closure, specimen thickness, plane stress and plane strain, 3D elastoplastic finite element method, plastic lateral contraction at the fatigue crack tip

## Abstract

Fatigue crack growth experiments are performed using A7075-T6 compact tension (CT) specimens with various thicknesses *t* (1–21 mm). The stress intensity factor at the crack opening level K_op_ is measured, and the effects of *t* and the stress intensity factor range ΔK on K_op_ are investigated. In addition, the change in K_op_ value due to specimen surface removal is investigated. Furthermore, we clarify that the radius of curvature of the leading edge of the fatigue crack decreases as *t* becomes thinner. Using the three-dimensional elastoplastic finite element method, the amount of plastic lateral contraction (depression depth *d*) at the crack tip after fatigue loading is calculated quantitatively. The following main experimental results are obtained: In the region where ΔK is 5 MPam^1/2^ or higher, the rate of fatigue crack growth da/dN at a constant ΔK value increases as *t* increases from 1 to 11 mm. The da/dN between *t* = 11 and 21 mm is the same. Meanwhile, in the region where ΔK is less than 5 MPam^1/2^, the effect of *t* on da/dN is not observed. The effects of *t* and ΔK on the da/dN–ΔK relationship are considered physically and quantitatively based on *d*.

## 1. Introduction

The importance of fatigue crack closure (FCC) on fatigue crack growth (FCG) behavior is acknowledged by many researchers. In many materials, even if the minimum stress intensity factor (K_min_) is on the tension side, a crack will not open unless the K value reaches the opening stress intensity factor K_op_ of the crack (K_op_ > K_min_). Therefore, K_op_ values must be evaluated well to accurately predict FCG behavior. Elber introduced the important concept of FCC [1,2] in the 1970s. He conducted FCG experiments using aluminum alloy 2024-T3 under a constant stress ratio—R value and demonstrated that K_op_ increased with the stress intensity factor width ΔK (= K_max_ − K_min_). Here, K_max_ is the maximum stress intensity factor. He proposed that the plastic stretch (plastic wake) occurring behind the crack tip contributed to the increase in K_op_. Similar results were observed in other low- and medium-strength aluminum alloys [3], and this type of FCC behavior is called plasticity-induced fatigue crack closure (PIFCC). After the initial study by Elber, other types of FCCs were discovered, such as roughness-induced fatigue crack closure (hereinafter RIFCC) [4,5,6], oxide-induced crack closure, and corrosion product-induced crack closure [7,8,9,10]. In RIFCC, unlike PIFCC, K_op_ levels do not change significantly or remain constant with increasing ΔK.

In a review paper regarding the PIFCC phenomenon [11], Schivje indicated that the K_op_ level in the plane-stress region (specimen surface region) was higher than that in the plane-strain region (inside the specimen). Ishihara et al. [12] conducted FCG experiments using A6061 and carbon steel S25C. They reported that A6061 exhibited PIFCC behavior, and the slope of K_op_–ΔK was almost 0.5, whereas carbon steel S25C exhibited RIFCC behavior and the slope of K_op_–ΔK was almost 0. In addition, they measured the fracture surface roughness (R_a_) near the specimen surface and specimen inside as a function of ΔK to specifically observe the interaction between the specimen surface and the specimen inside. Their results showed that the R_a_ near the specimen surface increased with ΔK, but the R_a_ inside the specimen decreased. Newman et al. reported that FCG behavior in aluminum alloy 7075-T651 (*t* = 5.7 mm) showed PIFCC behavior [13]. Furthermore, Matos et al. [14] investigated the effect of specimen thickness *t* on K_op_ using aluminum alloy 6082-T6. They observed that the K_op_ for thin specimens was higher than that for thick specimens. In addition, Camas et al. [15] reported that the leading edge of the fatigue crack had a curvature, and that this curvature was affected by the specimen thickness.

The conclusion of a review that we conducted on previous studies focusing on PIFCC for aluminum alloys indicates that knowledge regarding the effects of *t* and ΔK on FCG and FCC behaviors of aluminum alloys is limited to qualitative understanding. In particular, few studies have been carried out on the effect of ΔK on PIFCC.

In this study, FCG and K_op_ behaviors of A7075-T6 were investigated experimentally using CT specimens with various thicknesses. Subsequently, the effect of *t* on K_op_ and the range of ΔK at which the effect of *t* occurs were clarified. The effect of *t* on the leading edge shape of the crack was experimentally investigated; furthermore, the effect of the leading edge shape of the crack on K_op_ is discussed herein. In addition, the PIFCC behavior involving plastic deformation was significantly affected by the ΔK level. It is assumed that the effect of RIFCC appeared in the low ΔK region. Therefore, we analyzed the ΔK value at which the transition from PIFCC to RIFCC occurred. The effects of *t* and ΔK on K_op_ were quantitatively and physically considered by analyzing the plastic lateral contraction (depression depth *d*) of the crack tip via the three-dimensional (3D) elastoplastic finite element method (FEM).

## 2. Materials, Specimens, and Experimental Methods

### 2.1. Material/Specimen

The test material was aluminum alloy 7075-T6 (hereinafter A7075-T6). Its chemical composition and mechanical properties are shown in Table 1 and Table 2, respectively. A tensile test was performed according to JIS standards, and two specimens were used. The difference between the results of the two tests was not large; thus, the average value of these is presented in Table 2. The stress–strain curve obtained from the tensile test is shown in Figure 1. The stress–strain curve was approximated by Ramberg–Osgood power law, and a strain-hardening exponent of 0.036 was obtained.

FCG experiments were performed using American society for testing and materials (ASTM) standard compact tension (CT) specimens; their shapes and dimensions are shown in Figure 2. FCG experiments and K_op_ measurements were performed using four CT specimens with *t* = 1, 6, 11, and 21 mm, and with specimen width (*W*) = 57.2 mm fixed. FCG experiment was performed using one or two CT specimens for one specimen thickness, *t*. Such a method is employed in a usual FCG experiment using CT specimens. Then, the relationship between the rate of fatigue crack growth da/dN and ΔK was measured by performing a constant ΔK experiment, a ΔK increasing experiment, and a ΔK decreasing experiment. As for the measurement of K_op_, during a constant ΔK test, the values were measured multiple times (up to 4 times) at an arbitrary crack length. In the FCG experiments, experiments using three-point bending [16] have been also used relatively often.

### 2.2. FCG Experiment and K_op_ Measurement

In a laboratory environment, an FCG experiment was performed using a hydraulic servo fatigue-testing machine under a stress ratio R of 0.1 and a frequency of 15 Hz. The replica method [17] was used to measure the crack length. The FCG experiment was interrupted after every predetermined number of load cycles. Subsequently, methyl acetate was dropped on the specimen surface, and an acetyl cellulose film was quickly attached to the specimen surface to obtain a replica of the specimen surface. The crack length was measured by observing the replica at a 100× to 200× magnification using an optical microscope. Equation (1) [18] was used to calculate the stress intensity factor K, where *P* is the applied load, and *t* is the specimen thickness.
(1)$$\begin{array}{c} K=\frac{P}{t\sqrt{W}}\frac{2+\alpha }{{\left(1-\alpha \right)}^{3/2}}\left(0.886+4.64\alpha -13.32\alpha^{2}+14.72\alpha^{3}-5.6\alpha^{4}\right),\\ \alpha =\alpha /W,\;\alpha \geq 0.2, \end{array}$$

The elastic compliance method [19] was used to measure K_op_. To improve the measurement accuracy, a strain gage was attached in front of the crack tip, and the strain in the direction perpendicular to the crack propagation direction was measured. To increase the measurement sensitivity of K_op_, the signal of a load cell (PCD-320A; Kyowa Electronic Instruments Co., LTD, Tokyo, Japan) and the signal of the strain gage (PCD-300B; Kyowa Electronic Instruments Co., LTD, Tokyo Japan) were input to the subtraction circuit. The opening load of the crack was measured using the elastic compliance method, and the value was used to calculate K_op_. For details regarding the experimental method, please refer to a previous report [12].

### 2.3. Specimen Surface Removal Experiment

A CT specimen of *t* = 6 mm was used for the specimen surface removal experiment. First, a fatigue crack was propagated under a constant ΔK value (8.0 MPam^1/2^), and the FCG velocity and K_op_ value were measured each time. When the K_op_ value stabilized, the FCG experiment was stopped, and the specimen was removed from the fatigue testing machine. Subsequently, 0.5 mm on one side of the specimen (1 mm on both sides) was removed using an electric discharge machine (Tape Cut Model; FANUC Corporation, Chicago, IL, USA). At that time, sufficient care was taken such that processing strain was not introduced in the specimen. Subsequently, the specimen was attached to the testing machine again, and the FCG experiment was restarted. K_op_ was measured for each constant crack growth amount in this manner.

### 2.4. Measurement of the Curvature on the Front Edge of the Crack

To observe the shape of the leading edge of the fatigue crack inside the specimen, the specimen was removed from the testing machine when the ΔK value reached 12.0 MPa·m^1/2^. A red penetrant flaw-detection dye was infiltrated into the crack surface. Subsequently, the specimen was attached to the testing machine again, and the specimen was broken by applying a larger fatigue load. The fracture surface was observed using a scanning electron microscope to examine the leading edge shape of the crack. The observations above were performed on specimens with different *t*.

## 3. 3D Elastoplastic FEM

To quantitatively consider the effects of *t* and ΔK on K_op_, the amount of the plastic lateral contraction (concavity depth *d*) of the specimen was analyzed using 3D elastoplastic FEM. After the maximum load was applied to the CT specimen, the load was removed. The concavity depths, *d*, generated at the crack tip at that time were obtained using the FEM for various ΔK values with *t* values as a parameter. In the analysis, we assumed that the crack length *a* was constant at 15 mm (*α* = 0.26) and that the leading edge of the crack was straight (radius of curvature = ∞). In addition, in some of the calculations, an analysis was performed when the leading edge of the crack had a radius of curvature; subsequently, it was compared with a straight leading edge on the crack. Commercially available software [20] was used for the FEM analysis.

Examples of element division for the entire specimen and the vicinity of the crack tip are shown in Figure 3a,b, respectively, where *t* is 6 mm. A quarter of the CT specimen was analyzed, considering the symmetry of the specimen. Isoparametric elements were used for the FEM analysis; the total numbers of elements and nodes were 11,200 and 46,275, respectively. In addition, the minimum element size near the crack tip was set to approximately 1/75 of the plastic zone size at the crack tip [15]. For element division in the specimen thickness direction, the half thickness of the CT specimen was divided into 40 parts (minimum element size = 20 μm) in the vicinity of the crack tip to ensure calculation accuracy. With an increase in the distance from the crack tip, the numbers of element division in the specimen thickness direction were 20 (element size = 150 μm), 10 (element size = 300 μm), and 5 (element size = 600 μm); they were gradually reduced.

The material properties used in the analysis were a Young’s modulus of 70 GPa, a Poisson’s ratio of 0.3, and a yield strength of 510 MPa. Three-dimensional elastoplastic FEM analysis was performed on a material with a strain-hardening exponent of 0.036, and the plastic lateral depression depth *d* at the crack tip was calculated.

## 4. Experimental Results and Discussion

### 4.1. Relationships between FCG Rate da/dN and ΔK and between da/dN and ΔK_eff_


Figure 4 shows the relationship between da/dN and ΔK of A7075-T6 alloy on a log–log graph, where *a* is the crack length, and *N* is the number of stress cycles. In the figure, the experimental results for *t* = 1, 6, 11, 21 mm are shown. For comparison, the results for *t* = 5.7 mm obtained by Newman [13] and that for *t* = 10 mm by Jono et al. [21] are provided. As shown in the figure, the da/dN values for *t* = 6 and 11 mm in this study almost agreed with the values of *t* = 5.7 mm measured by Newman and *t* = 10 mm measured by Jono et al., respectively.

In the region where ΔK was 5 MPa·m^1/2^ or higher, da/dN increased as *t* increased from 1 to 21 mm at a constant ΔK value. Meanwhile, in the region where ΔK was 5 MPa·m^1/2^ or less, the da/dN value at each ΔK value did not depend on *t*; specifically, for ΔK between 3.5 and 5 MPa·m^1/2^, the da/dN values were the same for *t* = 1, 6, and 11 mm. For ΔK between 2.5 and 3.5 MPa·m^1/2^, the da/dN values for *t* = 5.7 mm (Newman) and *t* =10 mm (Jono et al.) were equal.

Figure 5 shows the relationship between da/dN and the effective stress intensity factor range ΔK_eff_ (= K_max_ − K_op_). Furthermore, the figure shows the result of the A7075-T651 CT specimen studied by Newman [13] and that of A7075-T6 alloy reported by Tokaji et al. [22]. As shown in the figure, the results of this study almost corresponded to the results of Newman [13] and Tokaji et al. [21]. Moreover, the effect of *t* was not observed in the da/dN–ΔK_eff_ relationship. Similar results were obtained by Ishihara et al. [12] using A6061 CT specimens and by Matos et al. [14] using A6082 CT specimens. Therefore, ΔK_eff_ was effective as the FCG driving force even when *t* changed [12,14].

When the da/dN–ΔK and da/dN–ΔK_eff_ relationships are plotted on a log–log graph, an inverted S-shape is shown [23]. In steel materials, excluding the vicinity of the threshold of the FCG (ΔK_th_) and unstable crack growth regions, the da/dN–ΔK relationship in the intermediate FCG region is shown as a straight line; hence, the Paris law applies. However, in aluminum alloys, as shown in Figure 4 and Figure 5, the intermediate FCG region can be further classified into three regions [20]; ① a low-rate region: da/dN = 10^−9^ to 10^8^ m/c; ② a medium-rate region: da/dN = 10^−8^ to 10^−7^ m/c; ③ a high-rate region: da/dN = 10^−7^ to 2 × 10^6^ m/c. Jono et al. [20] reported the fracture surface of the ZK141-T7 aluminum alloy as follows: in the low velocity region of ①, the fracture surface was featureless and comprised flat and uneven fracture surfaces; in the medium velocity region of ②, it was primarily a flat area (plateau area) surrounded by steps; in the high velocity region of ③, striations appeared.

The authors conducted an FCG experiment using an A7075-T6 CT specimen with a thickness of 1 mm and observed striations on the fracture surface at ΔK = 10 MPam^1/2^ and a da/dN of approximately 10^7^ m/c. Furthermore, it was observed that the striation interval near the specimen surface was narrower than that inside the specimen [24]. In the high-velocity region of ③ (ΔK is 5 MPam^1/2^ or higher), where the effect of *t* was clearly observed on da/dN (as evident from the ductile striation on the fracture surface), plastic deformation at the crack tip was considered to be an important factor for FCG.

### 4.2. Specimen Surface Removal Experiment and ΔK_eff_


To investigate the interaction between the surface and the inside of the specimen affecting the FCG behavior, a surface removal experiment was performed when the fatigue crack was growing. Before and after removing the specimen surface, the change in K_op_ with the crack growth amount Δ*a* was measured. Figure 6 shows the measurement results. As shown in the figure, the K_op_ value decreased from 3 to 1.75 MPam^1/2^ due to the removal of the specimen surface, and the K_op_ value gradually returned to the original value as the crack grew. The K_op_ value on the specimen surface was higher than that inside the specimen by approximately 1 MPam^1/2^. Hence, the effect of the specimen surface on the K_op_ value was significant. A similar finding has been reported by the authors of [12].

### 4.3. Effects of t and ΔK on K_op_


#### 4.3.1. K_op_–ΔK Relationship

Figure 7a shows the K_op_–ΔK relationship for the A7075-T6 alloy used in this study. In the figure, the experimental results for various *t* from 1 to 21 mm are plotted. Additionally, the figure shows Newman’s result [13] for CT specimens with a thickness of 5.7 mm, which correspond to the thickness of 6 mm in this experiment. When ΔK was 5 MPam^1/2^ or higher, K_op_ increased with ΔK, and the degree of increase was affected by *t*. The K_op_–ΔK relationships for each *t* were approximated by the straight lines; the slopes of the straight lines were 0.58, 0.32, and 0.2 for *t* of 1, 6, and 11 and 21 mm, respectively, and hence decreased with increasing *t*. Meanwhile, when ΔK was less than 5 MPam^1/2^, the K_op_ value was not affected by *t*. More specifically, the K_op_ values for *t* of 5.7, 11, and 21 mm were the same, and the slope of the straight line was not affected by *t*.

The K_op_ value of *t* = 6 mm at ΔK = 3.5 MPam^1/2^ in the present study was larger by approximately 1 MPam^1/2^ than that of *t* = 5.7 mm. As this value was larger than the experimental results of other thickness (*t* = 1, 5.7, 11, and 21 mm) in the low ΔK region, it was assumed that the value contained an error. Hence, in this study, a K_op_ value of 5.7 mm [13] was used.

In order to confirm the certainty and generality of the experimental results for A7075-T6 in this study, this experimental result (Figure 7a) is compared with the experimental results of Bao et al. [3] and Matos et al. [14]. The K_op_–ΔK relationships shown below were newly created for this study using the relationships da/dN–ΔK, and da/dN–ΔK_eff_ obtained by Bao and Matos [3,14].

Figure 7b shows the K_op_–ΔK relationship measured by Bao using A6061 CT specimens with a thicknesses of 0.35 and 6.35 mm [3]. From the figure, similar to A7075-T6, when ΔK was 5 MPam^1/2^ or higher, the K_op_ value of the specimen with a *t* of 0.35 mm was larger than that with *t* = 6.35 mm. Furthermore, the slope of the straight line for *t* = 0.35 mm was approximately 0.5, which was larger than 0.33 for the specimen with *t* = 6.35 mm. The values of these slopes were almost identical to those of A7075-T6. Meanwhile, when ΔK was 5 MPam^1/2^ or less, the K_op_ values for the specimens with *t* = 0.35 and 6.35 mm were almost equal.

Figure 7c shows the K_op_–ΔK relationship (R = 0.1) of the A6082-T6 alloy measured by Matos et al. [14]. K_op_ was measured using a backside strain gage. This figure was recreated by the author based on data obtained directly from the original paper. For comparison, the results (Figure 7a) for A7075-T6 of the present study are shown by broken lines in the figure. From the figure, for a constant ΔK value, K_op_ decreased with increasing *t* from 3 to 10 mm; however, the K_op_ values for the 10 and 25 mm specimens differed only slightly. The slopes of the straight lines shown in the figure showed values similar to the result (broken lines) of A7075-T6 in the figure. As discussed above, although the materials differed, the K_op_–ΔK relationships of A7075-T6 (this study), A6061 [3], and A6082-T6 [14] exhibited similar characteristics.

Elber [1] conducted FCG experiments using A2023-T3 at various R ratios and discovered an empirical formula of ΔK_eff_/ΔK = 0.5 + 0.4R. Considering the relation of ΔK_eff_ = K_max_−K_op_, K_op_ is expressed as the following equation (Equation (2)) [25]:(2)$$K_{\mathrm{op}}=\left\{1/\left(1-R\right)-0.5-0.4R\right\}\Delta K,$$

Substituting R = 0.1 of this study into Equation (2) yields a K_op_/ΔK value of 0.57. This value corresponds to a value of 0.58 when *t* is thin (plane-stress PIFCC) in the present study; however, its value is 0.2 when *t* is thick (plane-strain PIFCC). 

PIFCC depends largely on the ΔK level as it is largely dominated by plastic deformation in the specimen surface region. Therefore, strictly speaking, the condition without the influence of PIFCC is considered to be represented by K_op_/ΔK = 0 [12]. The physical meanings of the values of 0.57 and 0.2 for K_op_/ΔK, obtained in the present study, are currently unknown. Further research is needed in the future on this point.

#### 4.3.2. K_op_–*t* Relationship

Figure 8 shows the K_op_–*t* relationship with the ΔK value as a parameter. Experimental data are not available for the values of K_op_ at ΔK = 4.5 and 5 MPa m^1/2^ in the figure. Hence, these K_op_ values were estimated by extrapolating the straight lines in Figure 7a to the low ΔK region. In the figure, the result at ΔK = 12 MPam^1/2^ (R = 0.1) obtained by Matos [14] and that for *t* = 5.7 mm obtained by Newman [13] are shown.

As shown in the figure, at ΔK = 12 MPam^1/2^, the K_op_ value decreased from 6 to 2.5 MPam^1/2^ as *t* increased from 1 to 15 mm. As the ΔK value decreased to less than 12 MPam^1/2^, the effect of *t* on the decreasing rate of K_op_ became less. At ΔK = 4.5 and 5 MPam^1/2^, the *t* dependence of K_op_ almost disappeared. Meanwhile, when *t* was 11 to 15 mm or more, K_op_ was constant without depending on *t*. This was because the plane-strain PIFCC [26] dominated when *t* became thicker than a certain value, and K_op_ was not affected by *t*. This result is consistent with the fact that the 21 mm slope of K_op_–ΔK with *t* = 11 is of the same value, 0.2 (Figure 7a), which also corresponds with the result [14] of Matos et al. in Figure 7c.

The results of the FCC experiments can be summarized as shown in Table 3.

① When ΔK was 5 MPam^1/2^ or higher and *t* was 11 to 15 mm or less, PIFCC under the interaction of plane stress and plane strain occurred. As *t* became thinner, the plane-stress PIFCC dominated, the K_op_ value increased, and the slope of K_op_–ΔK approached 0.58. On the contrary, as *t* became thicker, the plane-strain PIFCC became more significant, and the slope of K_op_–ΔK approached 0.2. ② When ΔK was 5 MPam^1/2^ or higher and *t* was 11 to 15 mm or more, the plane-strain PIFCC dominated, the *t* dependence of K_op_ disappeared, and the slope of K_op_–ΔK approached 0.2. ③ When ΔK was 5 MPam^1/2^ or less, it was assumed that RIFCC occurred instead of PIFCC because the plastic deformation was small. As the deformation was small, no interaction occurred between the plane-stress and plane-strain regions, and the slope of K_op_–ΔK was 0.2 regardless of *t*.

In the ASTM standard [27], the following equation is recommended as a condition for *t* to obtain a plane-strain fracture toughness value K_IC_ using a CT specimen. Here, *σ*_Y_ is the yield strength of the material:(3)$$t\geq 2.5\left(K_{\mathrm{IC}}/\sigma_{Y}\right),$$

Osaki et al. [28] reported that the K_IC_ and *σ*_Y_ values of A7075-T6 were 31.2 MPam^1/2^ and 486 MPa for SL (Short transverse–Longitudinal) specimens, respectively, and 43.6 MPam^1/2^ and 548 MPa for LS (Longitudinal–Short transverse) specimens, respectively. By substituting these values into Equation (3), the minimum values of *t* for obtaining K_IC_ were calculated; the values of *t* = 10.3 and 15.8 mm were obtained for SL and LS specimens, respectively. Compared with these values, the minimum value of *t* (11 to 15 mm) for the plane-strain PIFCC obtained in this study is considered a reasonable value.

### 4.4. Shape of the Front Edge of the Crack 

Figure 9 shows the fracture surfaces of specimens with various *t* (1, 6, 11, and 21 mm). The leading edge shape of the crack was revealed by dyeing it red with a penetrant flaw dye. As shown in the figure, the leading edge shape of the crack was not straight but curved, i.e., the crack grew faster in the specimen interior than on the specimen surface. Based on the specimen surface removal experiment shown in Figure 6, the K_op_ near the specimen surface was higher than that at the specimen interior. The FCG driving force ΔK_eff_ (= K_max_ − K_op_) near the specimen surface was lower than that at the specimen interior. Therefore, the FCG velocity near the specimen surface area became lower than that at the specimen interior, and the leading edge of the crack exhibited an arcuate shape. Next, the radius of curvature *r* was measured by approximating the leading edge of the crack with an arc; the circular arc passed through both the center and surface of the specimen.

Figure 10 is a log–log graph showing the relationship between the radii of curvature *r* and *t*. As shown, for a specimen with a small *t*, *r* was small and increased with *t*.

## 5. Depression Depth *d* Generated at the Crack Tip

In order to quantitatively consider the effect of *t* on the K_op_–ΔK relationship (Figure 7) and the transition conditions for the ΔK value from RIFCC to PIFCC (Table 3), the depression depth *d* generated at the crack tip of the CT specimen was analyzed using 3D FEM. Analysis was performed for various ΔK values with *t* as a parameter. 

### 5.1. Effect of d on K_op_

Figure 11 shows the relationship between *d* and ΔK obtained using the 3D elastoplastic FEM for various *t* values. The curves in the figure were obtained by approximating the data with quadratic curves using the least-squares method.

As shown in the figure, at the region where ΔK was 5 MPam^1/2^ or higher (regions ① and ② in Table 3), *d* increased as *t* became thinner and ΔK increased; *d* can be considered as a physical quantity that comprehensively represents the interaction between the surface of the specimen (plane-stress deformation) and the specimen interior (plane-strain deformation). Therefore, when *t* was small, the constraint on the lateral contraction due to the specimen interior was small, and, hence, *d* increased. Furthermore, as ΔK increased, the out-of-plane plastic deformation at the crack tip increased, and, hence, *d* increased. Meanwhile, at the region where ΔK was less than 5 MPa m^1/2^ (region ③ in Table 3), the *d* value was small, i.e., approximately 0.05 µm, regardless of *t*. Hence, it was believed that RIFCC occurred instead of PIFCC because the plastic deformation necessary to cause PIFCC did not occur in this region. In a PIFCC, the amount of plastic stretch *ε* (plastic wedge) that occurs behind the crack tip and that at the crack-tip opening displacement are considered to be important factors [12]. It can be assumed that the material at the lateral contraction portion is supplied as the material for forming the plastic stretch *ε* behind the crack tip. In that case, it is assumed that as *d* becomes deeper, both *ε* and K_op_ increase.

### 5.2. Effect of the Leading Edge Shape of the Crack on the d Value

Figure 12 shows the relationship between *d* and ΔK obtained using FEM for *r* = 4.5 mm, the radius of curvature of the leading edge of the crack, and *r* = ∞ (straight edge). The FEM analysis conditions were *t* = 6 mm and *a* = 15 mm. The curves in the figure were obtained by approximating the analysis result with a quadratic curve. As shown in the figure, at a constant ΔK value, the value of *d* at *r* = 4.5 mm was larger than that at *r* = ∞. Moreover, this difference increased with ΔK.

Because K_op_ increases with *d*, the value of K_op_ for a curved edge with a radius of curvature of *r* = 4.5 mm, increases to larger than that for a straight edge. Camas et al. [15] investigated the effect of *r* on the plastic zone size in front of a crack using a 3D elastoplastic FEM and obtained results similar to those in this study.

The leading edge of the crack immediately after the start of the FCG experiment was a straight edge (*r* = ∞); however, as the crack grew, the leading edge shape of the crack gradually exhibited a curve (Figure 9). When the leading edge of the crack became curved, the value of K_op_ on the specimen surface increased to larger than that when it was straight; therefore, the change in the leading edge of the crack into an arc shape accelerated. Recently, FCG and K_op_ has been evaluated using FEM simulation [29,30]. To predict more accurate FCG and K_op_ behavior, an FCG simulation incorporating the shape change of the leading edge of the crack is necessary.

## 6. Conclusions

In this study, FCG experiments and K_op_ measurements were performed using A7075-T6 CT specimens with different thicknesses *t*. In addition, an FEM analysis was performed to supplement and quantitatively consider the experimental results. The effects of *t* and ΔK on FCG and K_op_ were studied, and the following conclusions were obtained:In the da/dN–ΔK relationship, in the region where ΔK was 5 MPam^1/2^ or higher, da/dN at a constant ΔK value increased as *t* increased from 1 to 11 mm. The da/dN between *t* = 11 and 21 mm was the same. Meanwhile, in the region where ΔK was less than 5 MPam^1/2^, the effect of *t* on da/dN was not observed. Furthermore, it was discovered that the relationship of da/dN–ΔK_eff_ was not affected by *t* and that ΔK_eff_ was an effective FCG driving force.When ΔK was 5 MPam^1/2^ or higher and *t* was 11 to 15 mm or less (region ①), the K_op_ value increased as *t* decreased. The slope of K_op_–ΔK was asymptotic to 0.58 as *t* became thinner. Conversely, as *t* thickened, the slope of K_op_–ΔK approached 0.2. When ΔK was 5 MPam^1/2^ or higher and *t* was 11 to 15 mm or more (region ②), plane-strain PIFCC dominated, and the slope of K_op_–ΔK approached 0.2, which did not depend on *t*. Meanwhile, in the region where ΔK was 5 MPam^1/2^ or lower (region ③), RIFCC occurred instead of PIFCC because the plastic deformation was small, and the slope of K_op_–ΔK became 0.2. Hence, ΔK = 5 MPam^1/2^ was the transition stress intensity factor from RIFCC to PIFCC.When the specimen surface was removed during the FCG, the K_op_ value decreased; subsequently, the FCG velocity returned gradually to the original value. The experimental results showed that the interaction between plane stress and plane strain was an important factor for FCG and K_op_.The leading edge shape of the fatigue crack showed an arc shape rather than a straight line. The radius of curvature *r* increased with *t*. Due to the interaction between the specimen surface and the specimen interior, ΔK_eff_ near the specimen surface became lower than that inside the specimen. Therefore, the FCG velocity near the specimen surface became lower than that inside the specimen, and the leading edge of the crack arcuated.According to the FEM analysis of the depression depth *d* in the region where ΔK was 5 MPam ^1/2^ or higher, the value of *d* increased as *t* decreased, and ΔK increased. When *t* was thin, the resistance against lateral contraction by the specimen inside was small, and, thus, *d* was large. When ΔK was large, *d* increased because the out-of-plane plastic deformation at the crack tip increased. Meanwhile, when ΔK was 5 MPa m^1/2^ or lower, the *d* value was small, i.e., approximately 0.05 μm, regardless of *t*; hence, RIFCC occurred instead of PIFCC. Therefore, the effects of *t* and ΔK on K_op_ (conclusion (2)) can be quantitatively and physically explained using *d*.

## Figures and Tables

**Figure 1 materials-14-00664-f001:**
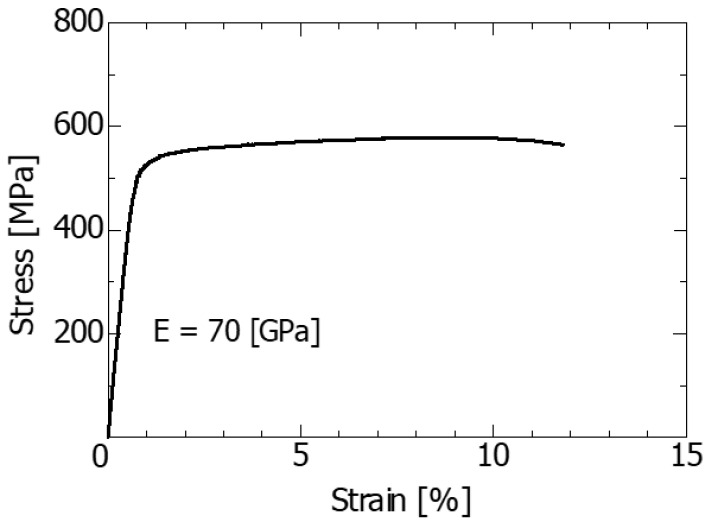
Stress−strain curve for the present material A7075−T6.

**Figure 2 materials-14-00664-f002:**
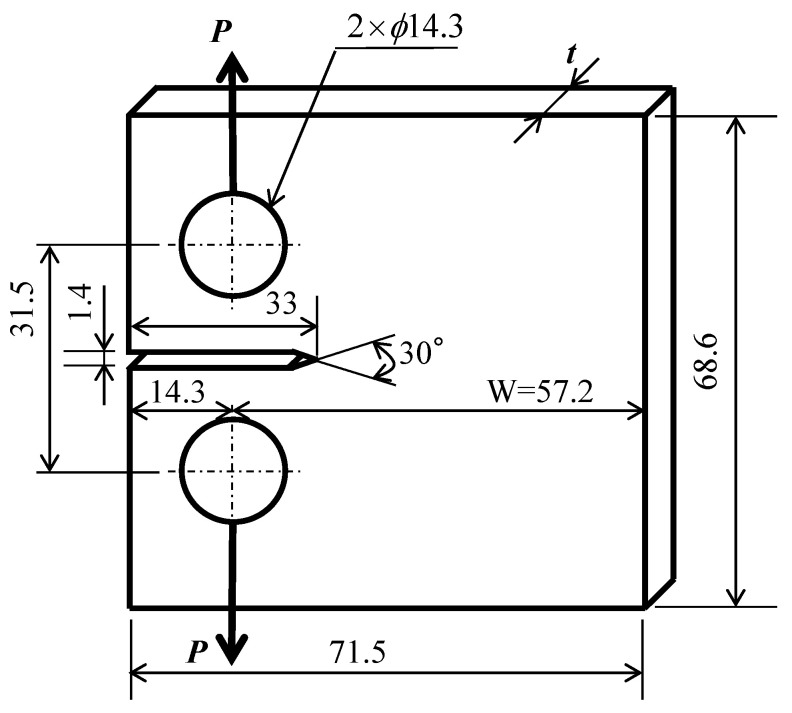
Shape and dimensions of ASTM standard compact tension (CT) specimens (*W* = 57.2 mm; thickness *t* = 1, 6, 11, and 21 mm).

**Figure 3 materials-14-00664-f003:**
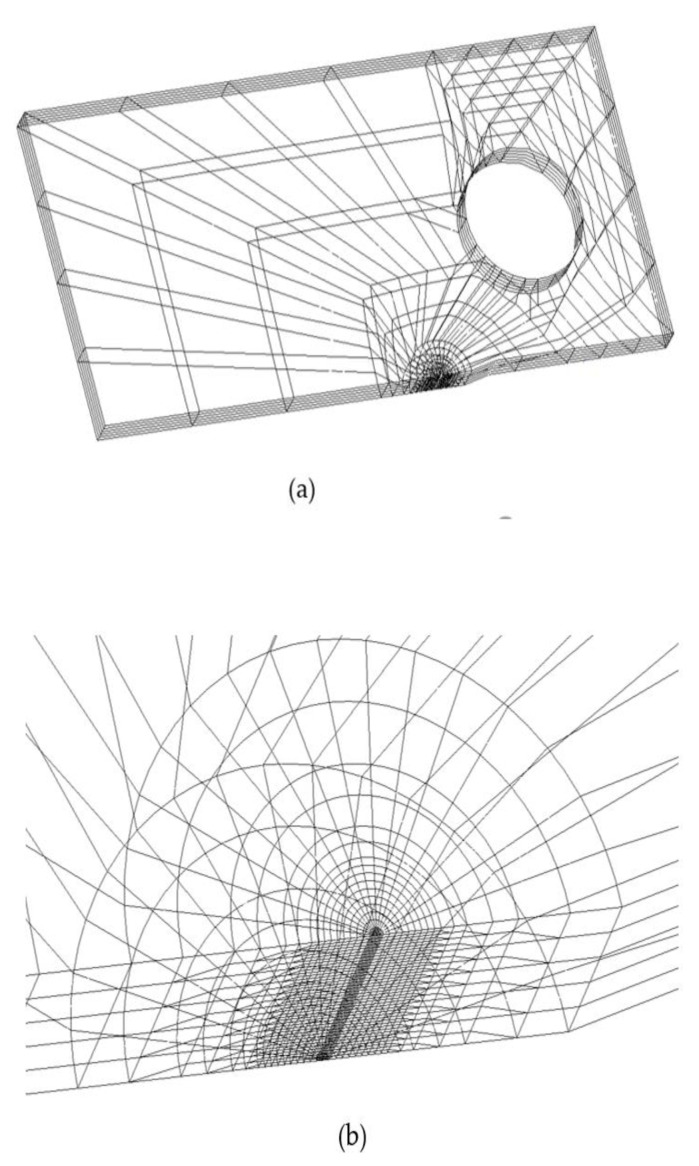
(**a**) Element division for a quarter of CT specimen; (**b**) enlarged view of the crack tip in Figure 3 (**a**).

**Figure 4 materials-14-00664-f004:**
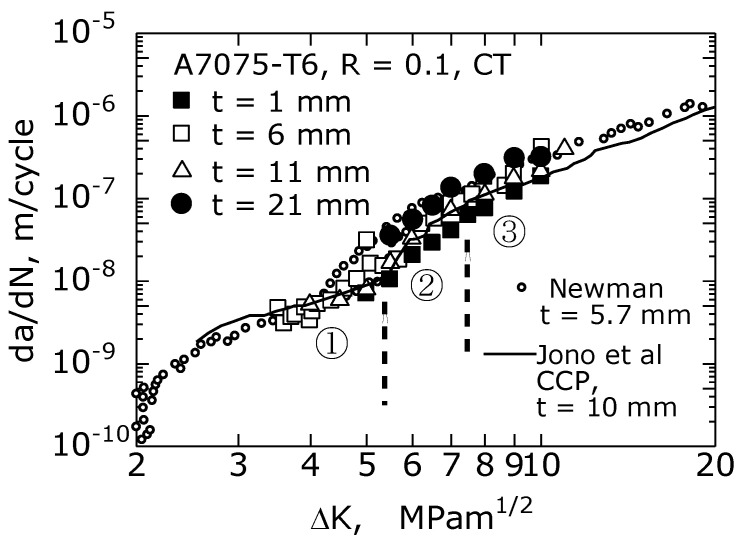
Fatigue crack growth–stress intensity factor range (da/dN–ΔK) relationship of A7075-T6 and the effect of *t* on the relationship, ① a low-rate region; ② a medium-rate region; ③ a high-rate region.

**Figure 5 materials-14-00664-f005:**
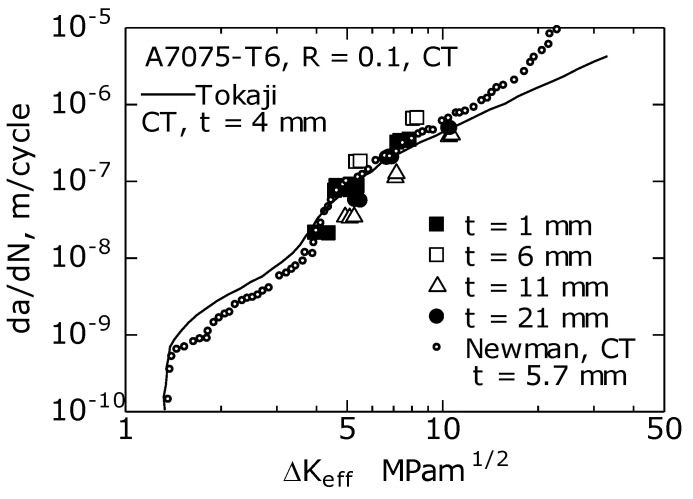
Relationship between da/dN and effective stress intensity factor range ΔK_eff_.

**Figure 6 materials-14-00664-f006:**
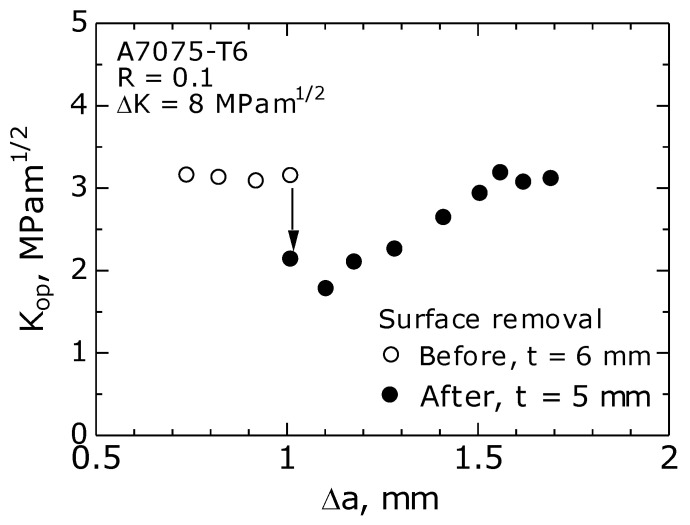
Specimen surface removal experiment (A7075-T6, ΔK = 8 MPa·m^1/2^, R = 0.1, *t* = 6 mm).

**Figure 7 materials-14-00664-f007:**
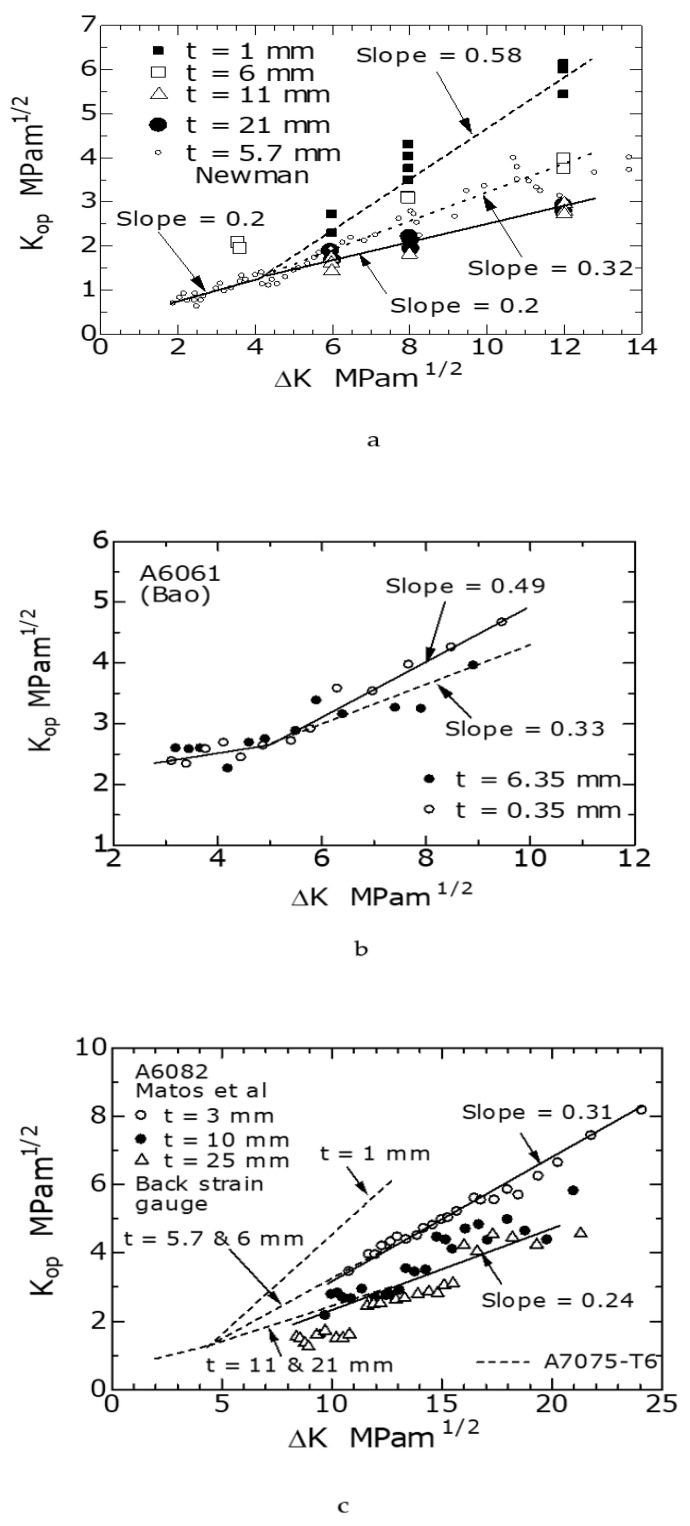
Effect of ΔK on the stress intensity factor at the crack opening level K_op_ for (**a**) A7075-T6; (**b**) A6061(Bao); (**c**) A6082-T6 (Matos et al.).

**Figure 8 materials-14-00664-f008:**
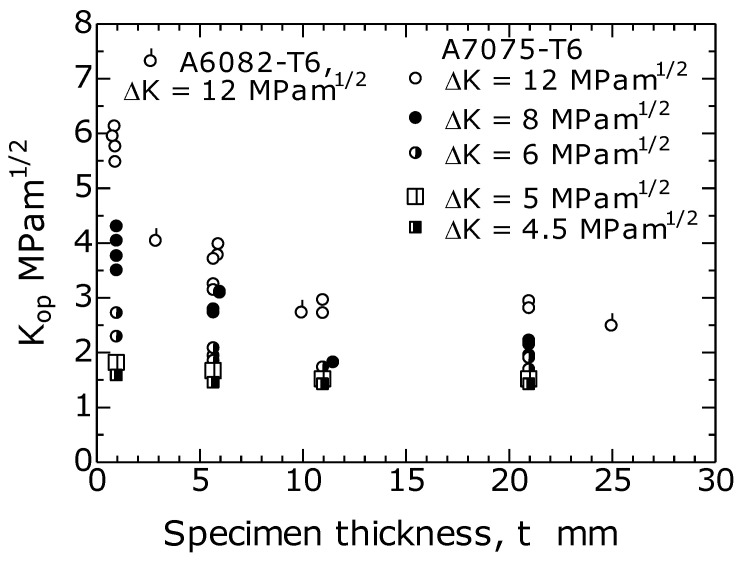
Effect of *t* on K_op_ (R = 0.1).

**Figure 9 materials-14-00664-f009:**
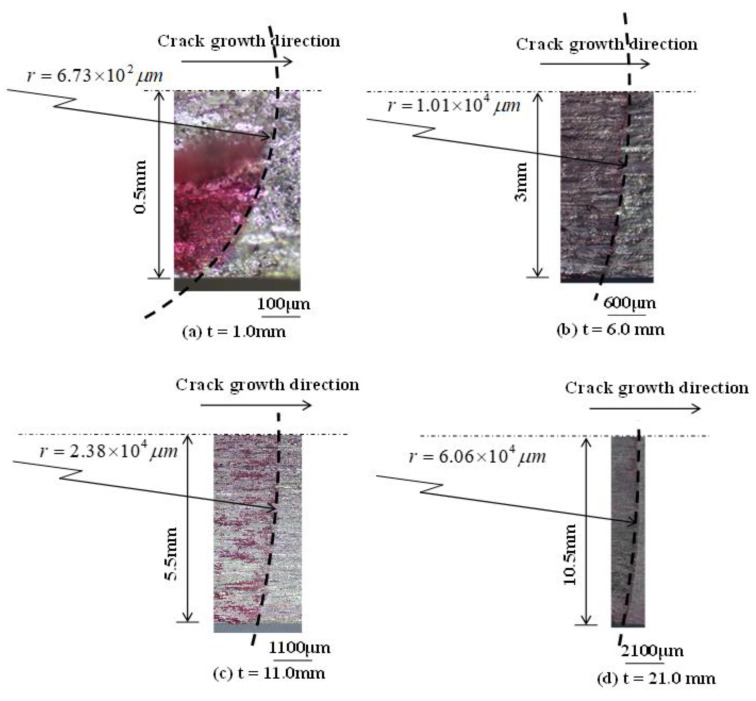
Leading edge shapes of the cracks of specimens with various thicknesses. ΔK = 12.0 MPam^1/2^, (**a**) *t* = 1 mm; (**b**) *t* = 6 mm; (**c**) *t* = 11 mm; (**d**) *t* = 21 mm.

**Figure 10 materials-14-00664-f010:**
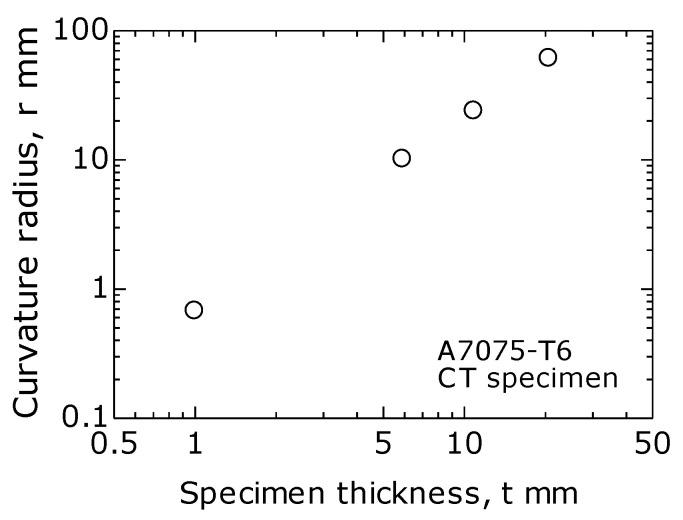
Relationship between curvature radius *r* and *t*. ΔK = 12 MPa·m^1/2^.

**Figure 11 materials-14-00664-f011:**
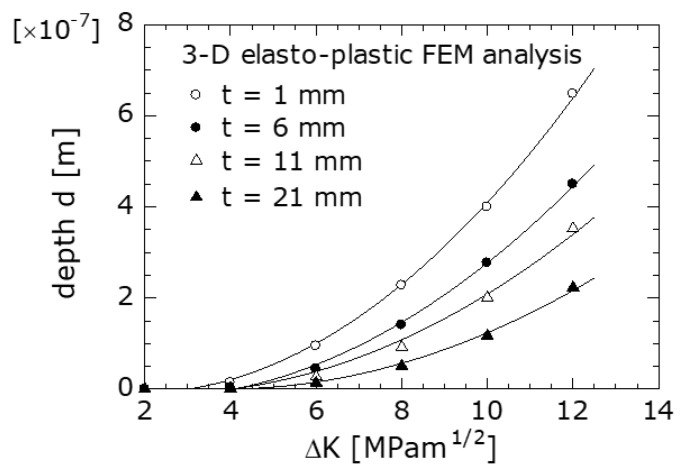
Relationship between depth *d* and ΔK.

**Figure 12 materials-14-00664-f012:**
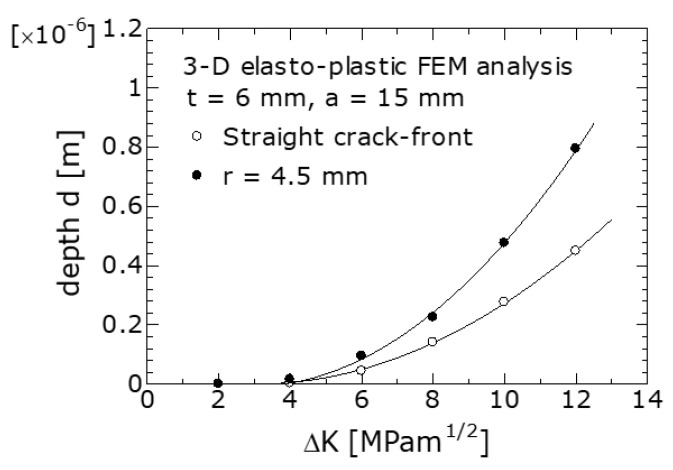
Effect of the radius of curvature *r* of the leading edge of the crack on the *d* value. *t* = 6 mm, crack length *a* = 15 mm.

**Table 1 materials-14-00664-t001:** Chemical composition of the test material (wt%).

Si	Fe	Cu	Mn	Mg	Cr	Zn	Ti	Al
0.40	0.50	1.60	0.30	2.50	0.24	5.50	0.20	Bal.

**Table 2 materials-14-00664-t002:** Mechanical properties of the test material.

Yield Strength	Tensile Strength	Young’s Modulus	Poisson’s Ratio	Elongation	Strain-Hardening Exponent
510 MPa	577 MPa	70 GPa	0.3	12%	0.036

**Table 3 materials-14-00664-t003:** Effects of *t* and ΔK on K_op_.

**Classification by ΔK**	③ ΔK < 5 MPa·m^1/2^	ΔK > 5 MPa·m^1/2^	
		① *t* < 11–15 mm	② *t* > 11–15 mm
**Classification of FCC**	RIFCC K_op_/ΔK = 0.2	PIFCC under interaction between plane stress and plane strain K_op_/ΔK = 0.2–0.58	Plane strain PIFCC K_op_/ΔK = 0.2
**Effect of *t***	No effect	Affected	No effect

## Data Availability

The data presented in this study are available on request from the corresponding author.

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
