# Peer review of "Effect of Specimen Thickness and Stress Intensity Factor Range on Plasticity-Induced Fatigue Crack Closure in A7075-T6 Alloy"

_materials, 2021, doi:10.3390/ma14030664_

Round 1

Reviewer 1 Report

Correct abstract, keywords. An interesting and comprehensive introduction, with a properly presented - but too succinctly in my opinion - review of the literature.

At the beginning, the paper lacks a list of symbols, markings and abbreviations. There must be nomenclature at manuscript. The authors consider the problems of material fatigue - many different quantities come up here and they need to be discussed. Nomenclature is essential in this type of papers.

In my opinion, in the paper Authors were missing the tensile curve - there is no given strain hardening exponent - although it can be determined based on API, SINTAP or FITNET procedures. But it is worthwhile for the authors to provide the laboratory obtained tensile curve - to write on which specimens the parameters given in Table 2 were determined, how many specimens there were and what was the scatter of the results - what result from the set is given in Table 2.

There is no information on the number of specimens used in the research program in the paper. The fact that values are given for four thicknesses, but in my opinion this is too scarce information. Fatigue tests generally have a very large spread and it seems to me that the authors should examine several specimens for one thickness to assess the spread of the experimental tests. Here, I expect explanations from the authors. As for the presentation of the results of the experimental research, their descriptions and discussions, I have no major reservations - they will disappear when the authors provide explanations to the questions that bother me.

As for the numerical calculations, the description of the model itself is too poor. Too little information on modeling. In the title of the paragraph, the authors indicate elasto-plastic analysis, and then in their paper they write "Considering the stress-strain curve of the test material, an elastic-perfectly plastic body was assumed." There is no logic here - the material used by the authors is a material with reinforcement - sigma_0 = 510MPa, sigma_m = 577MPa, according to the formula from FITNET procedures, the strengthening index N = 0.3 * (1-sigma_0 / sigma_m) - so the strain hardening exponent in the R-O law is equal to n = 1/N - substituting the data, we get n = 28. The material is therefore very weakly hardening and I do not think that such a material can be replaced with an elastic-perfectly plastic material. Please explain this - I don't quite like it at paper.

I guess figure 10 should be improved.

As for the analysis of the results, I have no major reservations - they will disappear once the Authors clarify the above doubts.

To sum up, I expect the authors to provide some clarifications and supplements in the manuscript. I consider it a minor revision / major revision. Then the paper should be submitted for re-evaluation.

Reviewer 2 Report

  1. Page 5 - the authors write about 3 regions in Figs 3 and 4. In Fig. 4, actually 3 regions are visible, unlike Fig. 3. While the increase in the velocity of cracking is a consequence of the developing crack. Only 2 regions are shown in Fig. 3?
  2. Fig. 8 - it is a pity that the samples in the drawings are not at the same (similar) magnification.
  3. It would also be worthwhile to quote the following paper: 1) Rozumek D., Faszynka S., Fatigue crack growth in 2017A-T4 alloy subjected to proportional bending with torsion. Frattura ed Integrità Strutturale 11(42), 2017, 23-29.

Reviewer 3 Report

Modern fracture propagation analysis is based on fracture mechanics concepts, based on the assumption that macroscopic fracture of a body is the result of crack propagation that occurs for various reasons. Taking into account the presence of cracks, fracture mechanics establishes a quantitative relationship between the stress acting on the body, the shape and size of the cracks, and the resistance of the material to the subcritical and supercritical development of these cracks. The present manuscript “Effect of specimen thickness and stress intensity factor range on plasticity-induced fatigue crack closure in A7075-T6 alloy” is of great theoretical and practical importance. The present article aimed on the study of various thicknesses effect on fatigue crack growth and crack opening stress intensity factor behaviors of A7075-T6 aluminium alloy. The article can be published after moderate revision. Some comments are follows:

  1. On table 1 the values must be adjusted in accordance with measurement accuracy
  2. Figure 9. Presented pictures are not clear and must be improved.

The present article “Effect of specimen thickness and stress intensity factor range on plasticity-induced fatigue crack closure in A7075-T6 alloy” by K. Masuda, S. Ishihara, N. Oguma can be accept after minor revision.

Round 2

Reviewer 1 Report

The all corrections suggested for the previous version of the paper have been included by the Authors in the new revised version of the paper. The paper put in by the Authors to improve the article meant that the paper has improved quality, is more readable and brings new elements. I recommend paper for publication in the Materials Journal.
